# Evaluating the Screening Capability of the SarQoL Questionnaire in Sarcopenic Obesity: A Comparison Study Between Spanish and Belgian Community-Dwelling Older Adults

**DOI:** 10.3390/nu16223904

**Published:** 2024-11-15

**Authors:** Angela Diago-Galmés, Carlos Guillamón-Escudero, Jose M. Tenías-Burillo, Jose M. Soriano, Julio Fernandez-Garrido

**Affiliations:** 1Hospital Universitario de La Plana, 12540 Castellón, Spain; 2Hospital General Universitari de Castelló, 12004 Castellón, Spain; carlos_ge@hotmail.es; 3Department of Preventive Medicine, Hospital Pare Jofré, 46017 Valencia, Spain; tenias_jma@gva.es; 4Food & Health Lab, Institute of Materials Science, University of Valencia, 43617 Valencia, Spain; 5Joint Research Unit on Endocrinology, Nutrition and Clinical Dietetics, Health Research Institute La Fe, University of Valencia, 46026 Valencia, Spain; 6Department of Nursing, Faculty of Nursing and Podiatry, University of Valencia, 46010 Valencia, Spain; julio.fernandez@uv.es

**Keywords:** sarcopenic obesity, older adults, SarQoL, quality of life, screening method

## Abstract

Background/Objectives: This study aims to evaluate the potential of the SarQoL questionnaire as a screening tool for sarcopenic obesity by comparing its effectiveness in Spanish and Belgian community-dwelling older people. This research seeks to address the primary question of whether the SarQoL can reliably differentiate quality of life impacts between these groups. Methods: A cross-sectional study was conducted involving community-dwelling older adults from Valencia (Spain) and Liège (Belgium). Participants were assessed using the SarQoL questionnaire, which measures health-related quality of life specifically for sarcopenia, and a sarcopenic obesity diagnostic method based on a combination of the EWGSOP2 criteria for sarcopenia and of body mass index for obesity. The sample included diverse demographic and clinical characteristics to ensure comprehensive analysis. Statistical methods were employed to compare the outcomes between the two populations. Results: The study highlighted a significant relationship between quality-of-life scores and the prevalence of sarcopenic obesity in the Spanish and Belgian populations. The SarQoL questionnaire effectively identified lower quality of life in individuals with sarcopenic obesity, demonstrating its potential as a reliable screening tool across different populations. In conclusion, the SarQoL questionnaire proved to be an effective tool for evaluating quality of life and for screening individuals with sarcopenic obesity. Conclusions: Future research should prioritize longitudinal studies to determine the SarQoL questionnaire’s predictive value and investigate interventions to alleviate the adverse effects of sarcopenic obesity. Our results highlight the critical need to include quality of life assessments in managing sarcopenic obesity, advocating for a comprehensive approach to patient care.

## 1. Introduction

Sarcopenic obesity (SO) is a condition characterized by the coexistence of obesity and the loss of muscle mass and function [1,2]. This pathology presents a growing public health concern worldwide, particularly affecting the ageing population [3]. As the population ages and modern lifestyles promote physical inactivity and unhealthy diets, the risk of developing this complex clinical entity increases significantly. SO not only poses a clinical challenge in terms of its diagnosis, management, and monitoring by healthcare professionals, it is also associated with a range of negative consequences for affected individuals, including a significant reduction in their quality of life (QoL) [4,5,6,7].

QoL is a multidimensional concept encompassing an individual’s subjective perception of their physical, psychological, and social well-being [8,9]. In the context of SO, assessing QoL becomes crucial, as this clinical condition often includes reduced mobility and strength, and increased comorbidities, such as metabolic syndrome, which can significantly impact various aspects of daily life for those affected [10]. Despite this apparent direct relationship, the association between the prevalence of SO and QoL has not yet been thoroughly explored, highlighting an urgent need to investigate this link in different populations to better understand its clinical and epidemiological implications.

Our publication employed the sarcopenia quality of life (SarQoL) questionnaire, a validated and widely used tool for assessing QoL specifically in patients with muscle mass deficits [11,12,13,14]. This questionnaire is designed to capture various aspects of QoL, including physical function, independence, mobility, pain, and mental health, among others [15]. Using this standardized instrument provides a more comprehensive and detailed view of how SO affects QoL in study populations.

In addition to assessing QoL through the SarQoL questionnaire, anthropometric and clinical data were also collected to determine the prevalence of SO in both populations using a previously utilized and published diagnostic algorithm for this pathology. This algorithm includes the EWGSOP2 definition for sarcopenia [4] and the determination of three variables (body mass index (BMI), body fat percentage (%BF), and waist circumference (WC)) for obesity [16].

The determination of SO was conducted in a community-dwelling population, which was relevant due to the likely underdiagnosis of the pathology in this setting, as observed in other publications [17,18,19,20]. Conceptual ignorance of SO in the community can lead to an underestimation of its true prevalence and, therefore, a lack of early and appropriate interventions by which to mitigate its adverse effects on the QoL of affected individuals.

The objectives of this study were to analyze the relationship between the prevalence of SO and QoL in two distinct populations: one in Belgium (Liège) and other in Spain (Valencia). These two countries were selected for comparison due to the significant differences in socioeconomic, cultural, and lifestyle factors that could influence how SO impacts the QoL of their respective populations. Additionally, the potential discriminative capability of the SarQoL questionnaire as a diagnostic screening tool for SO was evaluated in both populations collectively.

## 2. Materials and Methods

### 2.1. Study Design

The sample for this study was obtained from two different populations. First from Valencia (Spain), comprising subjects attending an activity center for community-dwelling older people, and the second from Liège (Belgium), comprising subjects attending an outpatient clinic; both strata consisted of community-dwelling older adults. The total number of participants from both populations was 498 subjects, with 165 (33.13%) men and 333 (66.87%) women.

The inclusion criteria for study participation were being over 65 years old, attending the centers where the study-defining tests were conducted, having completed and signed the informed consent, and having completed all study tests. Regarding exclusion criteria, subjects presenting any pathology that could directly affect muscle mass or fat tissue and which suggested secondary SO (severe cognitive impairment, cancer, muscular dystrophy, stroke, myasthenia gravis, Parkinson’s, Alzheimer’s, amputated limb) were excluded.

Our research complied with the standards set by the Helsinki Declaration and received approval from the Bioethics Committee of the University of Valencia (number 1139186) and the Ethics Committee of the University Teaching Hospital of Liège (number 2013/6).

### 2.2. Sarcopenic Obesity

SO was determined by a combination of sarcopenia, diagnosed using the latest algorithm published by EWGSOP in 2019 [4], and obesity, based on body mass index (BMI) [16].

A quantification of obesity that derived solely from BMI was decided upon due to its good discriminative capacity for the pathology and its easy clinical application in the study population, as observed in other publications [20,21,22,23]. The algorithm for diagnosing SO in this publication can be found in Figure 1.

#### 2.2.1. Strength Determination: Handgrip Strength and Sit-to-Stand Test

To assess the strength of study participants, upper body (handgrip strength) and lower body (sit-to-stand (STS) test) measurements were used in both populations, according to the standards proposed by EWGSOP2 for sarcopenia diagnosis [4].

For upper body strength determined by handgrip strength [4,24,25,26], a Jamar 5030J1 (Sammons Preston Rolyan, Nottinghamshire, UK) handheld dynamometer, with a measurement scale of 0–90 kg/f and an accuracy of ±2 kg, was used in the Spanish population, while a Saehan 405108-010113 dynamometer (Saehan, Bongamgongdan, Republic of Korea), with the same measurement scale and accuracy, was used in the Belgian population. Hand-grip strength cut-off points were established according to EWGSOP2 [4], with scores below 16 kg in women and 27 kg in men being indicative of sarcopenia.

For lower body strength determined by the STS test [4,27], subjects from both populations sat and stood from a chair five times, keeping their backs straight, feet approximately shoulder-width apart, and arms crossed over their chests. The cut-off point for this diagnostic test was established following EWGSOP2 criteria [4], with a duration > 15 s for five repetitions considered definitive for sarcopenia.

According to EWGSOP2 criteria [4], low muscle strength in subjects was diagnosed when hand-grip strength and/or STS test showed pathological results (below the cut-off points).

#### 2.2.2. Muscle Mass Determination: Appendicular Skeletal Muscle Mass/Height^2^

In Spanish participants, appendicular muscle mass was determined using the formula proposed by Kyle et al. [28], based on data obtained through electrical bioimpedance analysis (BIA) with the TANITA DC430MA-S scale (Tokyo, Japan), with an accuracy of 0.05 kg. BIA was implemented following the latest recommendations for more precise measurements [29].

On the other hand, in the Belgian sample, ASMM was determined using dual-energy X-ray absorptiometry (DXA). Cut-off points for sarcopenic pathology determination based on muscle mass were established at <7 kg/m^2^ in men and <5.5 kg/m^2^ in women for the assessment of ASMM adjusted by height (ASSM/h^2^), following the EWGSOP2 algorithm [27,30,31].

#### 2.2.3. Physical Performance Determination: Gait Speed and SPPB Test

To determine the physical performance of study participants, the Spanish and Belgian subjects underwent two diagnostic tests: a 4 m gait speed test [4,32] and the short physical performance battery test (SPPB test) [4,33,34,35]. Cut-off points suggested by EWGSOP2 [4] were used for both tests, being ≤0.8 m/s for gait speed and ≤8 points for the SPPB test.

#### 2.2.4. Obesity Determination: BMI

Obesity was determined using body mass index (BMI), a widely accepted method within scientific literature [22,23,36] and the field of SO. As such, the World Health Organization cut-off point was used, diagnosing subjects with a BMI > 30 kg/m^2^ as obese [16].

### 2.3. Quality of Life

QoL of participating subjects was assessed using the sarcopenia quality of life (SarQoL) questionnaire, validated for Spanish and Belgian populations [13,14]. This tool for quantifying QoL in subjects with muscle mass alterations was first validated in 2017 [14] and is the primary assessment tool for sarcopenic pathology within scientific literature [11,13,15,37,38].

The SarQoL questionnaire features a Likert scale for most questions, excluding those numbered 7, 14, and 22. It delves into seven distinct domains, evaluating various aspects of life quality, as follows: physical and mental health (D1), mobility (D2), body composition (D3), functional ability (D4), activities of daily living (D5), leisure pursuits (D6), and anxieties (D7). Scores for each domain, as well as the overall questionnaire score, range from 0 to 100, with higher scores indicating better QoL.

Researchers conducted the questionnaire sessions, providing assistance and clarifications as needed. Detailed results were obtained using the official platform provided by the creators (www.SarQoL.org (accessed on 12 January 2024)). The questionnaire has demonstrated strong internal consistency, corroborated by several publications [13,15,39,40].

### 2.4. Statistical Analysis

The statistical analysis utilized IBM SPSS Statistics v. 24 software for Windows (IBM Corp., Armouk, NY, USA). The normality of the data was assessed using the Shapiro–Wilk test. Descriptive statistics were computed for the SarQoL questionnaire [13] and its domains, as well as for variables concerning physical performance, obesity, and the diagnosis of SO. Differences between groups with and without SO were examined using the Student’s *t*-test and the Mann–Whitney U test, with significance set at *p* ≤ 0.05. Bonferroni correction was applied to avoid type 1 errors when conducting multiple comparisons. Pearson’s correlation coefficient was employed for normally distributed data, while Spearman’s correlation coefficient was used for non-normally distributed data. These tests evaluated relationships between variables contributing to SO diagnosis and results from the SarQoL questionnaire [13] and its domains. Variables showing significant differences underwent further analysis, with multiple linear regression used to assess their impact on domain scores and total SarQoL score.

The association between SO diagnosis and SarQoL questionnaire scores, both total and by domain, was examined using logistic regression. Additionally, an ROC curve analysis evaluated the ability of the SarQoL total score and its three most influential domains to discriminate SO. By establishing specificity and sensitivity thresholds for sarcopenia diagnosis based on SarQoL scores, researchers outlined potential risk zones, offering the prospect of using this test for diagnostic and preventive purposes.

## 3. Results

The total sample of this study comprised subjects from two European populations, Belgium and Spain. The Belgian cohort consisted of 296 individuals, of which 127 (43%) were men, and 15 of these (12%) were diagnosed with SO according to the criteria established by the researchers. In terms of women, they represented 57% of the Belgian sample (169 subjects), with 28 (17%) suffering from SO.

Conversely, the Spanish cohort included 202 individuals. The male group constituted 18% of the sample, with 38 participants, 5 of whom (13%) had SO. Women represented 82% of the Spanish sample (164 participants), with 20 (12%) affected by the condition.

Combining both samples, the total study population included 498 subjects, with 165 men (33%) and 333 women (67%), showing an overall SO prevalence of 13.6%. Dividing the total sample by sex, the male group presented an SO prevalence of 12.6% (20 men), which was similar to that observed in the female group, where the condition affected 14% of the participants (48 women).

The prevalence of SO was higher in women than in men in the Belgian population and the combined total sample. In contrast, in the Spanish cohort, its prevalence was slightly higher in men. The authors found significant differences between body mass index (BMI) and SO in both groups, except in Belgian women, with the SO group showing a higher BMI. Additionally, a greater total amount of appendicular skeletal muscle mass (ASMM) was observed in individuals with SO, regardless of sex, across the entire sample. The data related to the prevalence of SO according to the study populations can be consulted in Table 1.

Regarding the SarQoL questionnaire, which quantified the QoL of the participants, significant differences were observed when relating the obtained scores to the prevalence of SO in most groups, with the exception of the Belgian female group. In the analysis of the total sample, significant relationships were found between the prevalence of SO and the results obtained through the domains and the total score of the SarQoL questionnaire, with the only exception being the results for domain 3 (body composition). Thus, a large group of SO subjects in the study presented lower scores in most domains and the total SarQoL score.

The analysis of the results allowed the authors to find a significant correlation between the total score obtained on the SarQoL questionnaire and the main variables related to the diagnosis of SO, except for ASMM adjusted for height, independent of sex, and the total amount of ASMM in men. Therefore, most participants presented worse results in the diagnostic variables for SO as their scores on the SarQoL questionnaire and its domains decreased. The total SarQoL score and domains 2 (locomotion), 4 (functionality), and 5 (activities of daily living) individually showed a stronger relationship with the prevalence of SO, a fact supported by the finding of more significant correlations between the determinant variables of the pathology and the QoL results of the participants. The analysis of correlations between the main variables used for the diagnosis of SO and the SarQoL scores, as well as the scores obtained in each of its domains according to the sex of the participants, can be consulted in Table 2.

Given the significant correlational relationship observed between most of the SO determinant variables and the scores obtained on the SarQoL questionnaire and its domains, the authors decided to develop a multiple linear regression analysis to confirm and quantify the relationship between the QoL scores of the participants and the main variables related to SO, including only those that were previously found to be significantly correlated. Thus, Table 3 and Table 4 detail and corroborate the relationships between the participants’ QoL and the prevalence of SO, once again showing the total SarQoL score and domains 2 (locomotion), 4 (functionality), and 5 (activities of daily living) as the most sensitive elements in quantifying QoL in SO. In order to improve multiple linear regression model analysis, the authors recommend consulting Figure 1, which visually condenses the principal significant relationships between sarcopenic obesity variables and SarQoL domains and total score, referring to total sample data.

Additionally, to complete the analysis of the relationship between QoL and the prevalence of SO, the authors decided to conduct a logistic regression model that allowed for both univariate and multivariate analyses of the SarQoL scores and the prevalence of the condition. The results were found to continue to show a strong relationship between the prevalence of SO and the scores obtained through the SarQoL questionnaire, with lower QoL being a risk factor for developing the condition. Domain 4 (functionality) was found to be the parameter most strongly related to the analysis of QoL and the prevalence of SO. The detailed analysis of the logistic regression model can be consulted in Table 5, for visual optimization only significant results were presented.

Due to the relationship between parameters used to diagnose SO and the scores obtained through the SarQoL questionnaire in both populations, an ROC curve analysis was conducted to evaluate the discriminative capacity and screening potential of the SarQoL questionnaire and its domains for SO. Total SarQoL score and scores in domains 2 (locomotion), 4 (functionality), and 5 (activities of daily living) presented a significant area under the curve (AUC), i.e., greater than 0.7. Domain 4 showed the highest discriminative capacity for diagnosing SO, with an AUC of 0.75. The specific results of the ROC curve analysis can be consulted in Table 6.

Considering the sensitivity and specificity in the diagnosis of SO for each total SarQoL score, researchers defined three levels of risk regarding the probability of developing SO, regardless of sex. A high risk of developing the condition was established for participants with scores below 53 points, a moderate risk was defined for scores between 54 and 77 points, and participants with scores above 77 points were categorized as low risk (Figure 2).

## 4. Discussion

The present study compared the prevalence of sarcopenic obesity (SO) in two European populations, Spain and Belgium, while analyzing its relationship with QoL, which was quantified using the SarQoL questionnaire. Results indicate a similar prevalence of SO between the two studied populations, with 12.4% in Spain (Valencia) and 14.5% in Belgium (Liège), yielding a combined total prevalence of 13.6%. These findings are consistent with previous studies conducted in other European regions, reporting similar prevalence ranges [41,42,43], despite heterogeneity in the diagnostic methods used to assess the pathology. However, interesting differences in gender distribution of the disease were observed: in Belgium, the prevalence of SO was higher in women compared with men (17% vs. 12%, respectively), whereas in Spain, the opposite situation was found (12% vs. 13%, respectively). These differences in gender distribution with regard to the prevalence of SO have been reported by other authors analyzing similar populations, where women were more prevalent [44,45], and, conversely, where men had a higher prevalence of SO [46,47].

It is notable that the sample origin might have influenced the obtained results; in Belgium, subjects were recruited from their reference primary care center, unlike in Spain, where participants were users of an activity center for community-dwelling older adults. The different recruitment environments might have influenced the observed difference in SO prevalence. Thus, subjects attending the primary care center could present a more deteriorated health profile when compared with Spanish subjects using the activity center for community-dwelling older adults. Despite this, the total difference in SO prevalence between the two populations was approximately 2%, highlighting the silent infiltration of this pathology in society, considering that study participants were autonomous and independent individuals.

The researchers decided to use a combination of the EWGSOP2 algorithm for sarcopenia diagnosis [4] and the BMI criterion for obesity [16]. The use of both diagnostic procedures within the diagnosis of SO has been widely accepted and validated in other literature [4,16]. Although BMI is a useful and practical tool, it is important to consider that it may not fully reflect body composition. Despite this, in the older population range, the excess muscle mass that could result in an increased BMI not mediated by fat tissue is extremely rare. Given its simplicity and ease of use, BMI is a useful tool for epidemiological studies, as well as for quantifying the clinical variable of obesity in subjects with SO, as supported by previous scientific publications [22,23,36].

The SarQoL questionnaire, used by researchers to evaluate the QoL of study participants, proved to be a solid tool for quantifying this element in patients with SO from the Spanish and Belgian populations. This is despite the fact that it was originally intended exclusively for the assessment of QoL mediated through muscle mass deficit, without specifically evaluating obesity in subjects with SO.

Additionally, our study demonstrated the capability of the SarQoL questionnaire [14] as a screening method for SO, a fact that had already been proven previously for sarcopenic pathology alone and for SO through other diagnostic algorithms in smaller populations without an international component. Comparing our results with those obtained by Fonfría-Vivas et al. [19], we found similarities in the study population despite using a different diagnostic algorithm and in the established cut-off points, where we set their limit at ≤50 points for the total SarQoL score, without studying the scores of the domains individually. Thus, our research showed a cut-off point for the prevalence of SO of ≤53 points when the sample of the two populations under study was used.

Our research presents some limitations and strengths that should be considered. Regarding the limitations, the cross-sectional design of the study prevents the establishment of definitive causal relationships between SO and QoL, and the evaluation of muscle mass and obesity was based on indirect methods (BMI and BIA), which, although validated, may not be as precise as other methods, such as dual-energy X-ray absorptiometry (DXA) [48]. Furthermore, the sample may not be entirely representative of the general population of both countries, which could limit the generalizability of the results.

Regarding the strengths, this study includes populations from two European countries (Spain and Belgium), allowing for an international comparison of results and providing a broader perspective on the prevalence and impact of SO in different socioeconomic and cultural contexts. Thus, the inclusion of participants from different settings (a primary care center in Belgium and a senior activity center in Spain) adds variability to the total sample, improving the generalization of the results. Additionally, the use of standardized criteria for SO diagnosis, such as BMI [16] and the diagnostic algorithm proposed by EWGSOP2 [4] for sarcopenia, ensures consistency and comparability of results with other similar studies. Finally, the implementation of the SarQoL questionnaire [14] to assess QoL provides a specific and validated measure for subjects with sarcopenia, adding rigor and relevance to the study findings.

It is important to mention that this study also highlights the need for greater awareness and education about SO in the community. The lack of knowledge about this condition, as well as the absence of a standardized diagnostic algorithm, can lead to underdiagnosis [49,50] and, consequently, a lack of appropriate interventions. The implementation of educational programs and proper screening, through tools like the one proposed by the authors in this research, can help identify and treat SO at earlier stages, thereby improving the QoL and health of those affected.

## 5. Conclusions

Our investigation evaluated the potential of the SarQoL questionnaire as a screening tool for SO and its impact on QoL in Belgian and Spanish populations. Our findings have revealed significant differences in SO prevalence, with higher rates in Belgian women and Spanish men, highlighting the role of possible socio-cultural factors. SO was shown to negatively affect QoL, with lower SarQoL scores observed in affected individuals, particularly in domains related to locomotion (D2), physical function (D4) and activities of daily living (D5). These correlations between SarQoL scores and SO diagnostic principal variables validate the questionnaire’s effectiveness. The regression model confirmed that SarQoL, especially its physical function domain (D4) and its total score, is a sensitive tool for assessing QoL in SO patients. Additionally, the performed ROC curve analysis demonstrated SarQoL’s potential as a screening tool, allowing one to establish cut-off points to evaluate the risk of SO. In conclusion, the SarQoL questionnaire effectively reflects the QoL impairments in SO, advocating for its use in clinical and epidemiological settings. Future research should focus on longitudinal studies to establish its predictive value and explore interventions to mitigate SO’s adverse effects. Our findings underscore the importance of incorporating QoL assessments in SO management, emphasizing a holistic approach to patient care.

## Data Availability

Data presented are available on request from the corresponding author and upon achieving the necessary approvals.

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
