# Peer review of "Evaluating the Screening Capability of the SarQoL Questionnaire in Sarcopenic Obesity: A Comparison Study Between Spanish and Belgian Community-Dwelling Older Adults"

_nutrients, 2024, doi:10.3390/nu16223904_

Round 1
Reviewer 1 Report
Comments and Suggestions for Authors
This study is medologically well structured and makes important contributions to understanding the subject of sarcopenic obesity and quality of life. The theoretical framework is up-to-date and relevant.
I have identified some possible adjustments to improve the presentation of the final article:
1. I recommend removing the word potential from the title. It only increases the size of the title.
2. Also in the title, being told that it is a study involving the population of Belgium and Spain, perhaps creates the expectation of a population-based study. This is not the case. The way it is described in the abstract should be more realistic (lines 18 and 19 of the abstract).
3. I suggest that the data in table 3 be shown in graphs on a board with several graphs. Table 3 is not easily understood.
Author Response
Reviewer’s comment: This study is methodologically well structured and makes important contributions to understanding the subject of sarcopenic obesity and quality of life. The theoretical framework is up-to-date and relevant.
Author’s comment: First of all, authors would like to thank the reviewer’s job according to the different changing opportunities suggested for our research improvement. Thus, authors detail below the modifications done taking into account each of your specific comments made in your review. We hope it pleases you.
Reviewer’s comment: I recommend removing the word potential from the title. It only increases the size of the title.
Author’s comment: Thank you for your input, it was corrected.
Reviewer’s comment: Also in the title, being told that it is a study involving the population of Belgium and Spain, perhaps creates the expectation of a population-based study. This is not the case. The way it is described in the abstract should be more realistic (lines 18 and 19 of the abstract).
Author’s comment: Thank you for your comment. Authors have reformulated the final part of the title to avoid a possible misinterpretation of it. Also, with the intention of avoiding misinterpretations in the section of the abstract indicated by the reviewer, the text in lines 18-19 has been reformulated to be more adjusted to the sample under study.
Reviewer’s comment: I suggest that the data in table 3 be shown in graphs on a board with several graphs. Table 3 is not easily understood.
Author’s comment: Thank you for your comment. We understand Table 3 can be difficult to read due to the large amount of data it contains. In order to improve the understanding of this table, we have divided results in two tables, one with data from man and women and the other one with total sample data. Additionally, we also designed a new scheme that visually synthesizes the information contained in table 4, which one referred to total sample (Authors consider table 4 as the most relevant to be understood by readers because of its impact in this investigation).
Reviewer 2 Report
Comments and Suggestions for Authors
The manuscript describes a study aimed at evaluating the performance of a quality-of-life validated questionnaire in elderly subjects from to two independent populations, Belgian and Spanish to quantify the prevalence of sarcopenic obesity. The results indicate the existence of a significant correlation between the prevalence of sarcopenic obesity and quality-of-life scores. The questionnaire proved to be a reliable tool.
The study is well described, and the statistical methodology apparently correct. The number of citations is adequate. Figure 2 is acceptable.
A few remarks below are addressed to improve the quality of the manuscript:
Page 3, lines 110-116. It is not clear whether the condition of sarcopenia was established in the presence of HGS score below the cut-off points or STS above 15 seconds, or only in case of combined violation of both cut-off points.
Page 3, line 122. Did the authors mean that appendicular skeletal muscle mass was measured by DXA only in the Belgian population?
Page 9, Table 4. In the logistic regression analysis, it is not clear to me why not all odds ratios of the multivariate part were reported. I can guess that it was because they were not statistically significant or eliminated by some stepwise procedure. The authors should clarify this point.
Discussion. It is not clear why the authors emphasize that BMI does not fully reflect body composition (which I obviously agree with) if they collected DXA data used to quantify lean mass (page 3, line 122).
Minor:
Page 2, line 81. Defining the subjects participating in this study as “patients” is perhaps inappropriate.
Page 3, Figure 1. Please translate into English “absorciometria de rayos…” and “Bioimpedancia eléctrica”.
Page 3, line 111. Perhaps it is better to specify “hand-grip scores”.
Page 3, line 126. Please replace “y” with “and”.
Page 4, line 157. Perhaps it is worth adding “…to avoid type 1 errors when conducting multiple comparisons”.
Page 7, Table 2. Correct “Funcionality” to “Functionality”.
Author Response
Reviewer’s comment: The manuscript describes a study aimed at evaluating the performance of a quality-of-life validated questionnaire in elderly subjects from to two independent populations, Belgian and Spanish to quantify the prevalence of sarcopenic obesity. The results indicate the existence of a significant correlation between the prevalence of sarcopenic obesity and quality-of-life scores. The questionnaire proved to be a reliable tool.
The study is well described, and the statistical methodology apparently correct. The number of citations is adequate. Figure 2 is acceptable. The authors argue the need to adopt standardized measures to identify sarcopenic obesity.
Author’s comment: First of all, authors would like to thank the reviewer’s job according to the different changing opportunities suggested for our research improvement. Within this context, researchers have made several improvements with the aim of upgrading the information’s transmission of article’s topic and the quality in the content. Thus, authors detail below the modifications done taking into account each of your specific comments made in your review. We hope it pleases you.
Reviewer’s comment: Page 3, lines 110-116. It is not clear whether the condition of sarcopenia was established in the presence of HGS score below the cut-off points or STS above 15 seconds, or only in case of combined violation of both cut-off points.
Author’s comment: Thanks for your comment. We’ve added a little explanation between lines 127-129: “According to EWGSOP2 criteria [4], low muscle strength in subjects were diagnosed when hand-grip strength and/or STS test shown pathological results (below cutt-off points).”
Reviewer’s comment: Page 3, line 122. Did the authors mean that appendicular skeletal muscle mass was measured by DXA only in the Belgian population?
Author’s comment: The paragraph needed a clarification at the beginning. Your interpretation was correct and now it is clarified in line 137.
Reviewer’s comment: Page 9, Table 4. In the logistic regression analysis, it is not clear to me why not all odds ratios of the multivariate part were reported. I can guess that it was because they were not statistically significant or eliminated by some stepwise procedure. The authors should clarify this point.
Author’s comment: You’re right. To avoid possible misinterpretation, we’ve clarified it in Results section (line 253) and in Table 5 footnote (previous table 4).
Reviewer’s comment: Discussion. It is not clear why the authors emphasize that BMI does not fully reflect body composition (which I obviously agree with) if they collected DXA data used to quantify lean mass (page 3, line 122).
Author’s comment: Since there is a lack of information in Material and Methods section that refers to how muscle mass was quantified in the study, the confusion pointed out by the reviewer is understandable. With the corrections made, it can be seen that the sample of Spanish subjects quantified their muscle mass from BIA while the Belgian population obtained it through DXA. According to this difference in data collection, it was decided to use BMI as a diagnostic criterion for obesity in Sarcopenic Obesity because it is considered one of the most accessible tool for health care professionals around the world to diagnose obesity in the Sarcopenic Obesity context.
Reviewer’s comment: Page 2, line 81. Defining the subjects participating in this study as “patients” is perhaps inappropriate.
Author’s comment: Thanks for your comment. According to your comment, it has been replaced in the text.
Reviewer’s comment: Page 3, Figure 1. Please translate into English “absorciometria de rayos…” and “Bioimpedancia eléctrica”.
Author’s comment: It was a translation mistake. Thank you for your comment, now it is corrected.
Reviewer’s comment: Page 3, line 111. Perhaps it is better to specify “hand-grip scores”.
Author’s comment: Thanks for your comment. We’ve add “hand-grip strength” to the text.
Reviewer’s comment: Page 3, line 126. Please replace “y” with “and”.
Author’s comment: It was a translation mistake. Thank you for your comment, now it is corrected.
Reviewer’s comment: Page 4, line 157. Perhaps it is worth adding “…to avoid type 1 errors when conducting multiple comparisons”.
Author’s comment: Thanks for your comment. We’ve included your input in the text.
Reviewer’s comment: Page 7, Table 2. Correct “Funcionality” to “Functionality”.
Author’s comment: It has been corrected in table 2.
Round 2
Reviewer 2 Report
Comments and Suggestions for Authors
The authors have responded satisfactorily to my comments and the manuscript has improved. I have nothing else to ask.